# Friction Modelling for Tube Hydroforming Processes—A Numerical and Experimental Study with Different Viscosity Lubricants

**DOI:** 10.3390/ma15165655

**Published:** 2022-08-17

**Authors:** Lander Galdos, Javier Trinidad, Nagore Otegi, Carlos Garcia

**Affiliations:** Mondragon University, 20500 Arrasate-Mondragon, Spain

**Keywords:** hydroforming, friction, viscosity, finite element simulation

## Abstract

The final quality of sheet and tube metal–formed components strongly depends on the tribology and friction conditions between the tools and the material to be formed. Furthermore, it has been recently demonstrated that friction is the numerical input parameter that has the biggest effect in the numerical models used for feasibility studies and process design. For these reasons, industrial dedicated software packages have introduced friction laws which are dependent on sliding velocity, contact pressure and sometimes strain suffered by the sheet, and currently, temperature dependency is being implemented as it has also a major effect on friction. In this work, three lubricants having different viscosity have been characterized using the tube-sliding test. The final aim of the study is to fit friction laws that are contact pressure and sliding velocity dependent for their use in tube hydroforming modeling. The tests performed at various contact pressures and velocities have demonstrated that viscosity has a major effect on friction. Experimental hydroforming tests using the three different lubricants have corroborated the importance of the lubricant in the final forming of a triangular shape. The measurement of the axial forces and the final principal strains of the formed tubes have shown the importance of using advanced friction laws to properly model the hydroforming process using the finite element modeling.

## 1. Introduction

Tube hydroforming is a well-known technology for the production of lightweight, high strength and complex shape hollow components. A straight tube is first bent; if the final shape demands such an operation, then it is preformed with the closing of a two-sided tool and finally inflated or expanded by the use of pressurized media. The main advantages of the resulting component are the low weight and the high stiffness, which enables reducing the initial tube thickness to produce high strength lightweight components [1,2,3,4].

During the tube hydroforming operation, the tube is sealed by two axial cylinders. Lateral forces of these cylinders avoid the leakage of pressurized water–oil emulsion and additionally feed material from the guiding or feed zones to the expansion zones. One can understand this working principle as it is similar to the deep drawing, where material is fed from the blank-holder zone to the die cavity. At this stage, the reader may have already noticed that friction is very critical in tube hydroforming. A high friction makes the feeding of material to critical areas impossible and premature cracks appear during the process. As the contact pressure is high (normally in the range of 100–120 MPa) and the contact surface is large, the friction forces represent a big portion of the needed axial forces. The rest of the needed axial force is proportional to the internal pressure used in the process, which needs to be counteracted by the external push rods. Consequently, the correct estimation of friction forces is important for a good sizing of axial cylinders and an unfailing tool and process design.

According to Figure 1, three different zones can be defined in hydroforming: the so-called guiding or feed zones, the transition zones and the expansion zones. The later typically defines the component geometry and is machined to the nominal geometry of the component. As explained before, the guiding zones act as a reservoir of material. At this zone, the tube suffers axial compression together with through thickness compression caused by the internal hydrostatic pressure. The tool diameter in these areas is slightly larger than the nominal tube external diameter to ensure that despite the tube manufacturing fluctuations, the initial workpiece can be inserted in the tool cavity and closing of the tool does not plastify or pinch the tube. This initial gap, of about 0.25–0.5 mm in diameter, is eliminated when the tube is pressurized and reaches the critical expansion pressure.

For this reason, homogenous contact is only available at this area when the tube is slightly expanded in the circumferential direction and only after yielding of the tube occurs. After this expansion, the apparent contact pressure between the tube and the tool is nearly equal to the internal pressure of the pressurized media.

The transition zone is defined by smooth, sweep-like geometries and serve as a transition between the guiding zones and the desired component geometry. The radii of these areas suffer the highest contact pressures of the tool, as it happens in the entrance radii and punch radii of a stamping tool. The pressure that acts in the free areas of the tube are supported by these areas of the tool and thus the contact pressure is high at this transition features. The tube is typically axially compressed and circumferentially expanded in these zones. The compression level depends on the capacity of feeding material from the guiding zones to these transition zones, which is directly linked to friction.

Finally, the expansion zone defines the component geometry. Normally, plane strain or biaxial strain conditions are present in these areas of the tube. The contact only occurs at the final stages of the process and when the tube is almost completely inflated. At this stage, through thickness compression, stresses arise and friction becomes a critical aspect that governs the local thinning of these areas and the filling of small details that are present in the tool. Considerable surface expansions are typically observed in these areas, which considerably changes the tube outer surface micro-topography. For this reason, the tribological conditions in the expansion zones are different from the ones observed in the guiding zones. Surface expansion normally reduces the friction coefficient in comparison to non-deformed outer surfaces.

Finite element–assisted process design is well established in the automotive and aeronautical sectors for hydroformed components and is an effective step for a fast and reliable tool design. Traditionally, the coefficient of friction has been considered to remain constant during the simulation of a hydroforming process of a component. However, some stamping studies discussed the possibility of applying different constant coefficient of friction for each surface that is in contact with the sheet [5]. More recent tribological studies have revealed that the coefficient of friction is affected by several contact features. In this way, some authors have developed variable coefficient of friction models based on micro-scale contact behavior [6,7] or macro-scale [8,9,10] ones. In both cases, there is a generalized agreement that coefficient of friction is affected by the contact pressure and the sliding velocity [11]. As the contact pressure is increased, the irregular topography of the contact surfaces are subjected to a flattening of each asperities, so the contact geometry changes, resulting in a change of the coefficient of friction [12]. On the other hand, increasing the sliding velocity also decreases the friction of coefficient. According to the principle of tribology, under the boundary lubrication condition, the friction coefficient is mainly composed of fluid friction coefficient and solid friction coefficient. With increasing sliding velocity, the area of solid friction decreases gradually and leads to a decrease in the coefficient of solid friction. The local heating effect of the friction at the peak causes the viscosity of the lubricant to decrease and leads the coefficient of fluid friction to decrease. As a result, the friction coefficient decreases with increasing sliding speed [13].

The use of contact pressure and sliding velocity dependent friction laws is being progressively introduced in industrial stamping companies with the use of Triboform software recently acquired by Autoform. Recent industrial studies have proven that accuracy of numerical results increases when these new models are used [14,15,16,17]. Recently, studies have been published where temperature-dependent friction laws have been used in order to capture the heating up of the tools until the steady state production run is established [18,19]. Anyhow, none of these friction laws have been employed in tube hydroforming process modeling and their effect is unknown for the tribological conditions that are present in hydroforming (high contact pressures, variable sliding velocities and large surface expansions). In the current study, temperature influence is not considered. The process time in hydroforming is large, approximately of about 20 s, while the sliding velocities are small and the tube is in contact with internal pressurized water, which is able to absorb any heat up of the tube. For these reasons, it is assumed that temperature increase is negligible.

## 2. Friction Characterization in Hydroforming and Paper Objectives

Three kinds of tests have been used in hydroforming for friction coefficient measurements. The most common test used by several authors is the pressurized tube-sliding test, also called the push through test. The test was first presented by researchers from the Institute for Production Engineering (PtU) in Darmstadt to emulate the tribological conditions of the feeding zone [20,21]. During the test, a straight tube is pressurized until homogeneous contact between the tool and the outer surface of the tube occurs and is subsequently pushed by the use of one or two axial cylinders. The resulting friction force can be measured directly from the axial cylinders or indirectly from the reaction force that appears in the tool by the use of force transducers. The test can be performed using either a vertical or a horizontal configuration. In the vertical configuration, the tool is normally manufactured in a single part and the contact pressure is estimated from the internal pressure, making the hypothesis that the contact pressure is equal to the tube internal pressure. In the horizontal option, the die is split parallel to the tube horizontal axis and the contact pressure is calculated by measuring the vertical reaction of one of the half dies by using load sensors. In both cases, any friction between the dies and the moving punches is to be avoided and the dies must ideally lay on axial bearings to avoid any loss of transmitted forces. The Ohio State University and Paderborn University used the same principle to study the behavior of different lubricants [4,22,23] as it is a straightforward method to calculate the friction coefficient. Other similar and more recent studies are that of Refs. [24,25].

The second popular test is the tube-upsetting test and was initially created to estimate the friction coefficient in the forming or expansion zone of the tooling, although no biaxial surface expansion occurs during the experiment [22,23,26]. In the test, a straight tube is continuously upset in a closed die while pressurized by an internal pressure. For that, a constant relative velocity is set between the axial cylinders that close the tube. Therefore, tube-sliding and wall thickness thickening occurs at the same time. Due to friction forces the wall thickness increases non-uniformly being thicker in the moving punch side. The thickness distribution is used to estimate the friction coefficient by inverse numerical simulation and is therefore an indirect measurement system [27,28]. The test does not reflect the tribological conditions that are present in the forming zone and is more suitable for the feeding zone evaluation.

Thirdly, specific tests for the estimation of the friction coefficient in the expansion zone were developed. The same group that developed the tube-sliding test used a modified version of the method to evaluate the friction between the die and the tool after considerably expanding the tube using spacers [29,30]. Additionally, and in the same study, a special tool was constructed where four component load cells were used to monitor the forces in the transition radii and estimate the friction coefficients in these zones. A similar approach was used years before by Dohman et al. with the same aim [31]. These last methods were complicated to be reproduced and several researchers used a simpler methodology, the corner-filling test, to estimate the average friction coefficient in the forming zone. In [32], a pear-shaped tube expansion test was used to calculate the friction coefficient by inverse analysis. A similar approach was used in [25] to calculate the friction coefficient by using three different expansion shapes, including a square section die. A recent work used a square section tool equipped with three sensors [33] in order to calculate the average friction coefficient out of the tube expansion versus time curves measured in the corner areas. In [34], an industrial frame hydroforming process was used to estimate the coefficient friction using inverse simulation techniques.

All in all, and unlike in stamping, very few studies have been found where pressure and sliding velocity dependent friction laws have been used to numerically model the hydroforming processes by finite element modeling. Only Groche et al. used an adaptive friction model to estimate the process feasibility of a T-Shape component in [35]. The developed friction law was normal pressure, sliding velocity and surface expansion ratio dependent. Wall thickness distribution and dome height were measured experimentally and compared with the numerical results obtained with the adaptive friction law and with a conventional constant friction coefficient of 0.04. The later value was the result of a numerical optimization of friction coefficient with respect to wall thickness and dome height of the real experimental component. The study showed that an adaptive friction model provides more accurate results than a constant friction model. In the specific case study, only one relevant thickening area was predicted by the constant friction model while the adaptive model predicted two heavily thickened areas, like in the real component.

For the abovementioned reasons, the final objective of this work is to analyze the influence the variable friction laws have on the final numerical results of hydroforming. For this purpose, three different lubricants have been tested using the pressurized tube-sliding test. The viscosities of lubricants have been selected to cover a wide range of hydroforming applications. Low viscosity lubricants are used in the automotive sector to facilitate the final cleaning of the components while high viscosity ones are used in the aeronautical sector to enable the forming of very complex components in a single shot. Some researchers proved that the friction coefficient is very dependent on the lubricant viscosity and performed experimental tests using different sliding velocities and punch radii using flat sheets and relatively low apparent contact pressures [36,37]. Similar analysis has been performed in this paper for hydroforming applications using two low viscosity and a very high viscosity mineral oils.

Tube-sliding tests have been performed using different internal pressures and tube pushing velocities. These experimental results have allowed fitting variable friction models which are contact pressure and sliding velocity dependent. Surface micrographs have been obtained from tested tubes trying to identify the mechanisms that occurred during the tests in the most damaged contact areas and for different lubricant viscosities. Finally, the validity of the newly developed friction models has been evaluated by the hydroforming of a triangular shape component with a high perimeter expansion and the comparison of numerical and experimental results.

## 3. Experimental and Numerical Procedures

### 3.1. Tube and Tool Materials

Roll-formed and high frequency–welded tubes have been used in the study. The raw material of the tubes is a DC03 cold rolled and annealed mild steel of 1.35 mm of thickness. The mechanical properties of the tube material obtained from uniaxial tensile tests of samples cut from unfolded tube precuts are shown in Table 1. The outer diameter of the tube is 50 mm and the tubes are GI coated (pure zinc galvanized). The Ra roughness of the tube is approximately 0.5 µm (Sa = 0.65 ± 0.014).

The tube-sliding tool was manufactured using a 1.2379 tool steel, tempered to 55 HRc. The circular shape of the tool was machined using a ball-nose end mill of diameter 10 mm and an initial radial material stock of 0.25 mm starting from the hardened condition. The milling passes were always performed in the longitudinal direction of the tube to favor material feeding during the hydroforming process.

Surface topographies of the tube and the tool characterized by means of Sensofar S-NEOX, Barcelona, Spain, optical profilometer using interferometry technique are shown in Figure 2. Three measurements of 1666 × 1251 µm^2^ area were acquired with a 20XDI objective and the metrological software SensoMap Premium 7, Barcelona, Spain (Digital Surf) was used for data post-processing. The form was removed by means of second-polynomial fitting and representative 3D topographical parameters describing height (Sq), hybrid (Sdq, Sdr) and functional (Vmp, Vvv) properties were computed in the primary surface following ISO 25178-2:2012 standard (see Table 2).

### 3.2. Tested Lubricants

Three different mineral oils have been used in the study. The first two lubricants have low viscosity, 80 and 150 mm^2^/s respectively and they are typically used in the tube bending and forming of steel components. The third lubricant has a high viscosity, 1300 mm^2^/s, and it is typically used in extreme contact pressure applications and with critical components. All the lubricants have been produced by the same supplier, Rhenus, using similar formulations and are compatible and highly soluble with the water–oil emulsion used in the hydroforming facility. The properties of the three lubricants are summarized in Table 3.

### 3.3. Friction Characterization Technique—Tube-Sliding Tests

The tube-sliding test developed at Mondragon University and used in the present study is presented in Figure 3. Like in [4,24,25], only one axial cylinder is employed for pushing the tube sample after the desired pressure level has been reached. Unlike with two axial cylinders, the use of one cylinder avoids the need of using a minimum axial force in both sides of the tube, needed for guarantying the sealing of the tube inner cavity. On the other hand, and because the axial pushing element is guided in bearings, the friction force can be directly measured using only one load sensor.

The configuration of the test is horizontal and was installed in the hydroforming cell that is available in the Mondragon University. The vertical force is measured in the upper side of the tool by the use of a load sensor that was manufactured using a 50CrV4 spring steel hardened at 56 HRc (see Figure 4). A full Wheatstone bridge was used for the sensing of the new sensor and was calibrated using a universal compression machine. The upper side of the tool is guided using axial bearings to avoid any force loss caused by friction and acquire a precise measurement of the vertical reaction force. The hydraulic pressure intensifier and the axial cylinder are controlled using servo hydraulic valves of Bosch Rexroth and HNC100 electronic controllers.

Three different internal pressures—20, 40 and 60 MPa—have been used in the tests and two different sliding velocities have been evaluated, 0.5 mm/s and 5 mm/s. The tube yielding occurs approximately at 16 MPa and thus lower pressures are not testable with this tube material and thickness. The maximum pressure is just below the maximum hydroforming pressure that is needed to hydroform the selected component, 65 MPa. The sliding velocities were selected to cover the full range of typical velocities used for the hydroforming of different components. A quantity of 5 g/m^2^ of lubricant was manually applied in the tubes using a brush and verified by a precision scale. The tool was cleaned using acetone after each test and before a new experimental condition was tested.

A typical test result is shown in Figure 5. As it is observed, in the pressurization step the axial cylinder is static and only internal pressure increases progressively. At a certain pressure level, when tube yields, the pressure is transmitted in the form of vertical force to the vertical force sensor. After the pressure is in the desired test value, the tube is pushed by the axial cylinder, the sliding step starts and the axial force builds up. Note that, in this step, the vertical and axial force gradually increase. This effect was also observed by [4,23,24,38] and it is attributed to the local thickening of the tube in the push rod area and the progressive lubricant loss during the tube-sliding test.

The friction coefficient can be easily calculated from the axial and vertical forces. The measured axial feeding force, *F_a_*, for pressing the pushing rod is assumed to be equal to the friction force at the tube–die interface; thus, the axial feeding force can be expressed as:(1)Fa=μ·π·P·D·L 
where *P* is the contact pressure, *D* is the tool diameter and *L* is the tube length in contact with the die. Similarly, the vertical force, *F_v_*, can be calculated by multiplying the contact pressure with the projected contact surface of the tube–die interface:(2)Fv=P·D·L 

After the combination of the above formulae, the coefficient of friction is calculated as follows:(3)$$\mu =\frac{F_{a}}{\pi \cdot F_{v}}\;$$

As the friction coefficient is not totally constant during all the sliding step, the average value is calculated from the evaluated feeding distance as suggested by [23]. Three repetitions were made for all the conditions.

### 3.4. Hydroforming Finite Element Numerical Models

The hydroforming of a triangular shape component was analyzed by using the Abaqus Explicit simulation code. The selected component has a very high expansion ratio in the central area where the final perimeter of the component is approximately 40% higher than the initial tube (see Figure 6).

The tube was modeled using 13,680 S4R shell elements of 2 × 2 mm^2^ size. The material was modeled using a constant elastic modulus of 205 GPa and a Hill48 yield criteria together with an associated flow rule, which was defined by using the anisotropic coefficients shown in Table 1. The hardening of the steel was introduced by a Swift Hockett–Sherby law, with the parameters shown in Table 1. GOM Aramis small area technique was employed to guarantee a reliable extension of the flow curve, as explained in [39].

For the feasibility analysis, the Forming Limit Curve (FLC) was obtained using tube hydroforming tests. Bulge tests with lateral feeding were used to identify the left side of the FLC. The plane strain zone was characterized with bulge tests with no axial feeding. For the biaxial or right side of the FLC, specific elliptical hydroforming tests were performed, as explained in [40].

The tooling was considered rigid and was modeled with 10,395 R3D3 and R3D4 planar elements of 2.5 × 2.5 mm^2^. The contact between the tube and the rigid tool was defined using the penalty contact algorithm. For the friction law definition, the VFRIC user defined subroutine was used. This subroutine provides all the variables needed for the definition of the Filzek friction law, i.e., contact pressure and sliding velocity. The analytical friction laws were then defined for each lubricant using the parameters presented in Table 4. Furthermore, additional simulations were also performed using different constant friction coefficients of 0.05, 0.1 and 0.15. The results of these models were compared with the results obtained with the advanced friction laws and are presented in the next chapter.

## 4. Friction Characterization Results and Friction Modeling

The experimental results for all the lubricants and tested conditions are shown in Figure 7a. As it is observed the maximum coefficient of friction is approximately 0.18 for the lubricant having the lowest viscosity and when tested in the lowest internal pressure and sliding velocity. On the contrary, the minimum friction coefficient is observed for the highest viscosity lubricant but the maximum sliding velocity, although small difference is observed in comparison to the lowest velocity of the same condition. The results clearly show that friction coefficient decreases when the contact pressure, and the sliding velocity and the viscosity increase. The sliding velocity effect is less pronounced with the high viscosity lubricant and when contact pressure are high.

The experimental results have been fitted to a Filzek like friction law [8]. The Coulomb friction coefficient depends on the contact pressure and the sliding velocity as follows:(4)$$\mu =\mu_{0}\cdot {\left(\frac{P}{P_{0}}\right)}^{n-1}-k\cdot ln\left[\frac{\max\left(v_{rel,\;v_{0}}\right)}{v_{0}}\right]$$
where *µ*_0_ is the reference coefficient of friction at the reference contact pressure *P*_0_, *n* is the pressure exponent defined within the range 0 < *n* < 1, *k* is the velocity factor that corrects the effect of the sliding velocity and *V_rel_* and *V*_0_ are the sliding or relative velocity and the reference velocity respectively. The model parameters for the different lubricants are summarized in Table 4 and are plotted against the experimental friction values in Figure 7b–d.

Micrographs of the tested tubes have been obtained using optical microscopy. In all the cases, micrographs have been captured in the middle of the tube. For the tested tubes, the most damage areas of the sample were selected for their inspection in the same zone of the tube.

The initial outer surface micrograph of the tube before testing is shown in Figure 8a. The typical EDT texturing pattern is present in the tube surface in order to optimize the contact behavior of the tube during forming processes. After the tribological tests, the most damaged areas of the tube outer surfaces were selected for their evaluation. The 20 MPa micrographs showed that no change of external tube surface occurred when using the 1300 mm^2^/s viscosity lubricant. Flattening of initial asperities and plowing was observed for the low viscosity lubricants at the lowest internal pressure.

Figure 8b,c are surface micrographs of highly damaged zones of tubes tested using 60 MPa of internal pressure, a sliding velocity of 0.5 mm/s and the two low viscosity lubricants. As in 20 MPa (not showed to avoid duplicity), flattening of initial asperities and deep plowing traces are observed for these cases. Most probably, the tribological condition is in the limit between the mixed and boundary lubrication regimes. Figure 8d shows the surface micrograph of a tube tested using the high viscosity lubricant at the same conditions. Flattening of asperities is observed in this case, but strong evidence of plowing is not visible. This indicates that the higher viscosity lubricant is able to limit the solid-to-solid macro contacts and thus justifies the low friction values obtained for this lubricant in the tube-sliding tests.

## 5. Hydroforming Numerical and Experimental Results

### 5.1. Hydroforming Process Optimization

The hydroforming process was numerically optimized using the strategy shown in Figure 9. Initially, a linear increase of the pressure was defined in the software with no axial feeding of the tube ends, using a maximum pressure of 65 MPa (maximum pressure available at the hydroforming equipment).

After this initial simulation, the formability criteria, using the FLC, was checked to detect any failure that may occur in the process due to necking. Because forming of the component was not possible with the use of a linear pressure curve and with the absence of axial feeding, a simple strategy to define optimized process curves was applied (see Figure 9). First, the P1 pressure was defined from the initial simulation. This pressure value is the critical internal pressure to start inflating the tube in the central area of the component. Then, P2 and P3 were established by trial and error. P2 is the pressure that defines the end of the axial feeding (feeding from X1 to X2 position) of the tube ends at a relatively low pressure. A big axial feeding with high internal pressure causes very high friction forces and makes difficult the feeding of the material from the guiding zones to the critical forming areas of the component. The axial feeding was defined in the simulations as a boundary condition in the feeding direction. The final maximum pressure peak P3 was used to calibrate the component without the use of any extra axial feeding. At the end of the process, the pressure intensifier was driven to the original position to decrease any pressure from the tube inside, a stage which is not included in the simulations.

The maximum pressure of the intensifier and the maximum force of the axial feeding hydraulic cylinders, 25 t, made impossible the hydroforming of the component with the lubricants of low viscosity. Thus, the process was numerically optimized for the high viscosity lubricant. For this case, an axial feeding of 30 mm was defined for both cylinders. P1 was set to 25 MPa and the axial feeding ended at a pressure P2 of 40 MPa. The calibration of the tube was realized with a pressure of 65 MPa, the maximum pressure of the intensifier. The axial feeding rate was defined to be 0.6 mm/s (30 mm of material feeding in 50 s).

### 5.2. Tube Hydroforming Tests

The presented triangular shape components were hydroformed using the three lubricants and the process parameters optimized for the high viscosity one. Additionally, another tube was formed without using any lubrication. The tooling was machined using the same strategy as presented for the tube-sliding friction tests using a 5-axis milling machine and the lubricants were applied manually and verified by a precision scale (5 g/m^2^ of lubricant quantity). The tool was cleaned using acetone after each test and before a new experimental condition was tested.

As predicted by the numerical models, only the process where the high viscosity lubricant was used resulted in a sound component. The low viscosity lubricants and the process with no lubrication resulted in big cracks near the maximum expansion areas of the tube. As it is observed in Figure 10, the progressive increase of the lubricant viscosity helped for a better axial feeding of the material from the guiding zones to the expansion areas. This is clearly identifiable as the final length of the tubes is smaller when the viscosity increases.

### 5.3. Numerical and Experimental Results

In order to validate the new friction laws used in the numerical simulations and to quantify the influence of the viscosity in the final results, the principal major and minor strains (Forming Limit Diagrams), the final thinning in the central cross section of the formed component and the axial feeding forces for the three cases were experimentally measured and compared with the numerical results.

The final principal major and minor strains of the real hydroformed component (only the high viscosity case) were measured using the PHAST software (Company Geodelta, Delft, The Netherlands) and the photogrammetry technique. For this purpose, the metallic tubes were electro-chemically etched with a point grid with a spacing of 2 mm. The measured major strains are shown in Figure 11.

The forming limit diagrams of the real hydroformed component (only the high viscosity case) and the different numerical models are shown in Figure 12. The Forming Limit Diagram is used in sheet metal forming for predicting material failures by necking, being the forming limit curve (solid curve in the graphs) the limit for acceptable components. All the points above the limit curve will tear due to necking. The graph is a representation of the major and minor strains measured in the formed component sheet plane.

In Figure 12a, the experimental strain measurements are plotted together with the numerical results obtained with the three constant friction coefficient models (nu015, nu01 and nu005) and the new variable model corresponding to the high viscosity lubricant (Visc1300). In Figure 12b, the experimental strain measurements are plot together with the numerical results obtained with the three variable friction laws (Visc1300, Visc150, Visc80). As it is observed, the variable friction law fitted for the high viscosity lubricant shows a similar behavior of the simulation performed with constant friction of 0.05. In both cases, the principal strains of the tube at the end of the hydroforming are very similar although the most critical element of the variable viscosity case shows a slightly higher major strain. The lowest viscosity lubricants show similar trend to the model with constant friction coefficient of 0.15, a typical value that industry has been using for decades for hydroforming and stamping simulations. The low viscosity lubricants, which have quite similar viscosity values in comparison to the high viscosity case, show similar forming limit diagrams and principal strains, which reveals that higher viscosity changes are needed to considerably change the process behavior. In all cases, the variable friction laws tend to move left in the forming limit diagram in comparison to the constant friction value models. This leftwise movement may be caused by a lower friction value predicted by the variable friction law at high internal tube pressures and contact pressures. This makes easier the flow of the tube ends towards the hydroforming die, compressing the tube ends and main geometry and lowering minor principal strain (more negative minor strains). The experimental minor strain levels are more negative than the numerical predictions made by the variable friction law obtained for the high viscosity lubricant, which suggests that lower friction values are present in the real process than the ones predicted by the new Filzek model. This may be explained because the tube suffered more compressive strains during the feeding stage or during the expansion of the tube. This is further discussed in the conclusions chapter.

The component hydroformed with the high viscosity lubricant was EDM cut in the central area for the thickness measurement using a coordinate measuring machine. The thinning values in function of the measured angle are shown in Figure 13. The origin of the angle is shown in Figure 6. Note that the thickness at ±180 °C is 1.35 mm, the original thickness of the tube. This can be explained because the welded zone was intentionally located in that flat position. The heat-affected zone of the tube is more rigid than the base material, and thus, a negligible straining of the weld line is observed.

The thickness prediction of the model is quite reasonable in the most critical areas of the component, where the thinning is around 1.05 mm. However, the model underestimates the thickening suffered by the flat areas of the triangle, which are associated with the axial feeding of the material. This result is in good agreement with the forming limit diagram results, where a more compressive trend is also observed in the experimental case.

The axial forces needed to feed the material during the process are shown in Figure 14. The total numerical force was calculated by the sum of the axial force predicted by the numerical model in the nodes where the boundary condition was defined and the force needed to close the tube due to the internal pressure that is present at that instant of the simulation. Because the diameter expansion and thickening are negligible in the guiding zones, internal pressure was multiplied with the initial tube cross section area to compute the closing force value at every simulation step.

As it is observed in Figure 14, the predicted axial forces are in good agreement with the experimental measurements. Experimental forces are slightly lower than the numerical predictions, which is also in good agreement with the other experimental results. Lower forces may explain a faster thickening of the flat areas (thinning results) and the leftwise displacement of the experimental forming limit diagrams.

## 6. Discussion

Different authors have characterized the friction in tube hydroforming processes using the tube-sliding test, tube-upsetting tests and simulative hydroforming tests. Many different lubricants have been tested at different internal pressures and sliding velocities and their influence has been widely proved in the last two decades. In the present work, the tube-sliding method has been used to evaluate three different lubricants with three different viscosities, as it is the most straightforward and available test and is optimal for the feeding zone friction characterization, the most critical zone for the numerical simulation of hydroforming processes with axial feeding of material.

The experimental results showed that, the lower the viscosity, the higher the friction coefficient. Similarly, the friction increased when the contact pressure and the sliding velocity decreased. The experimental friction coefficients are in good agreement with other available publications. Friction values between 0.04 and 0.22 were reported in [34] for four different type of lubricants. The friction coefficients were obtained using inverse simulation and minimizing the error between the experimental and numerical thickness distribution predictions. Ngaile et al. predicted friction values between 0.075 and 0.125 for four different solid like lubricants [32]. Schuler recommended the use of friction coefficients between 0.25 and 0.38 for oils [38]. These friction coefficients are slightly higher than the ones obtained in this work although the same authors reported much lower friction values using the same testing facility and other lubricants [20,35]. Plancak et al. reported friction values between 0.004 and 0.07 by using the tube-upsetting method [23,26]. Same authors used the tube-sliding test to obtain friction coefficients of the guiding zones and reported values from 0.025 to 0.15 for mineral oils [22]. These last values are in good agreement with the friction coefficients obtained in the current work and indicate that the tube-upsetting test predicts much lower values than the tube-sliding test. This can be explained because the flattening of asperities is very pronounced in the tube-upsetting test (tube axial compression and hydrostatic pressure acting at the same time) and the local relative material movement is very small. Hwang et al. reported a friction value of 0.045 for a mineral oil using the tube-sliding test. The friction is quite low in comparison to our study, but soft aluminum tubes were tested and both the tube outer surface roughness and tool roughness were higher [24]. More recently, two research groups reported values of 0.22 in [27] and from 0.01 to 0.09 in [25] for different mineral oils.

The friction coefficient dependency with the contact pressure is similar for all the tested lubricants, although the low viscosity lubricants show a slightly smaller reduction slope in comparison to the high viscosity lubricant (see Figure 7a). In general, the sensitivity of the coefficient of friction to the sliding velocity is higher in the low viscosity lubricants. This may be explained because with these lubricants the tribological system is in the limit of boundary and mixed lubrication regimes and solid-to-solid contact is higher than in the high viscosity case. In these conditions, surface flattening may be higher than in the case of the high viscosity lubricant, which is near the hydrodynamic regime.

A Filzek-like analytical friction coefficient law has been successfully used for the fitting of the experimental data. In general, the model shows a good agreement with the experimental values and the minimum R-squared (R^2^) is 0.86. As explained before, the highest viscosity lubricant has the highest sensitivity to the contact pressure and this is reflected in a lower pressure exponent *n*. The medium lubricant presents the highest *k* and *n* coefficients, being very similar to the coefficients of lubricant1, the lowest viscosity lubricant. These coefficients indicate a lower sensitivity to the contact pressure and a higher sensitivity to the sliding velocity, although the general friction level is given by the reference friction coefficient *µ_0_*, which decreases with viscosity.

Regarding the outer surface micrographs, flattening of initial asperities is observed for all the lubricants at high contact pressures. Only the high viscosity lubricant at low pressures showed no flattening of initial EDT texture. Deep plowing traces are observed for the low viscosity lubricants indicating that lubrication regime is in mixed lubrication regime and not far from the boundary regime. On the contrary, the high viscosity lubricant tribosystem is in the mixed lubrication regime but some condition may indicate that we are near the hydrodynamic regime, especially at low pressures.

Finally, experimental hydroforming tests to form a triangular shape with high surface expansion were done. The tests clearly show the relevance of the lubricant viscosity in the material flow during the hydroforming and, thus, in the friction behavior. The highest viscosity lubricant enables the forming of a sound component while the lower viscosity lubricants are not able to reduce the friction value to an amount that allows the forming of the components with forces lower than 25 tons. As in the case of Groche et al. [35] the variable friction laws implemented in the Abaqus Explicit code show a different behavior in comparison to the constant friction coefficients. As explained before, the tendency when using advanced numerical models is to reach smaller and more negative minor strains, which means the friction coefficient decreases with the contact conditions of the different areas (contact pressure and sliding velocities). The experimental minor strain values are more in the left in the forming limit diagram than the numerical predictions using the advanced friction laws. These may be explained by two different hypothesis. Firstly, the contact pressures in the transition areas, the radii, are higher than the ones tested in the tube-sliding tests, which could still lower the friction coefficient at these regimes. Secondly, the tube expansion in the central areas of the triangle can cause a severe tube surface roughness flattening. Recent studies made by Hol et al. [6] suggest that friction coefficient decreases with surface expansion. A lower friction value in these areas would allow a higher material feeding which could explain the thickening of the tube in the flat areas, linked to the thickening results, and the left shifting of the forming limit diagram.

The prediction of the lateral axial forces is accurate when using advanced friction laws and this is relevant for the industry, as it permits a correct sizing of the hydraulic cylinders to be used in the hydroforming tool. Again, differences in predicted forces, which are higher than the experimental ones, may be explained by the hypothesis of higher contact pressures that are present in the transition radii, the influence of the surface expansion and roughness flattening or by a combination of both.

## 7. Conclusions

As in stamping simulation, friction is a highly influencing parameter in hydroforming processes. Tribological conditions depend on various input parameters and, among them, lubricant viscosity has been proved to be an important one.

In this paper, a friction variable friction law that accounts for sliding velocity, contact pressure and lubricant viscosity has been developed and implemented in the Abaqus Explicit software (Dassault Systèmes, Vélizy-Villacoublay, France). The simulation results have been compared to experimental tests and numerical results obtained using with numerical simulations where constant friction values have been used.

Numerical results are more reliable when variable friction laws are used. However, some small differences have been observed between the numerically calculated and experimentally measured principal strains. This is attributed to the fact that the new friction law does not account for the surface expansion and tube initial asperities flattening. For this reason, new tribological tests where the tube is first expanded and then tested using tube-sliding tests is recommended for future works.

## Figures and Tables

**Figure 1 materials-15-05655-f001:**
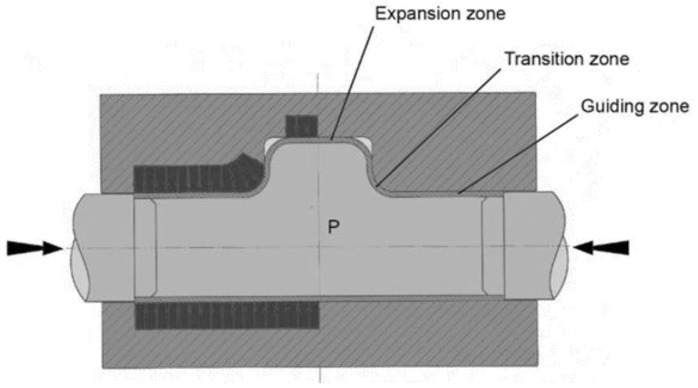
Different tool zones in tube hydroforming.

**Figure 2 materials-15-05655-f002:**
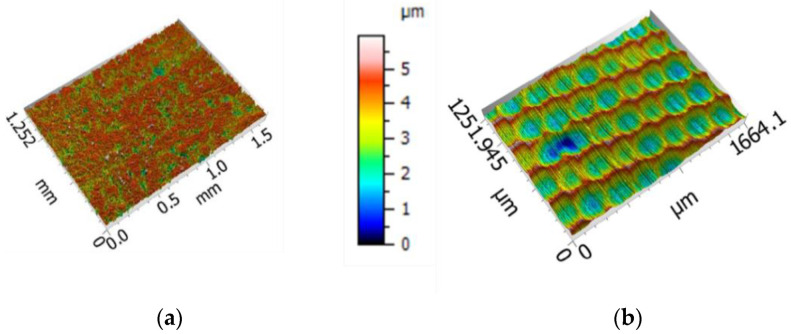
Surface topographies of (**a**) the tube and (**b**) the tool.

**Figure 3 materials-15-05655-f003:**
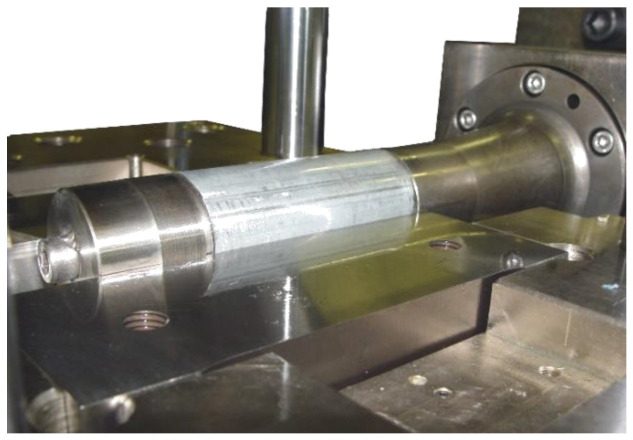
Tube-sliding test facility of Mondragon University.

**Figure 4 materials-15-05655-f004:**
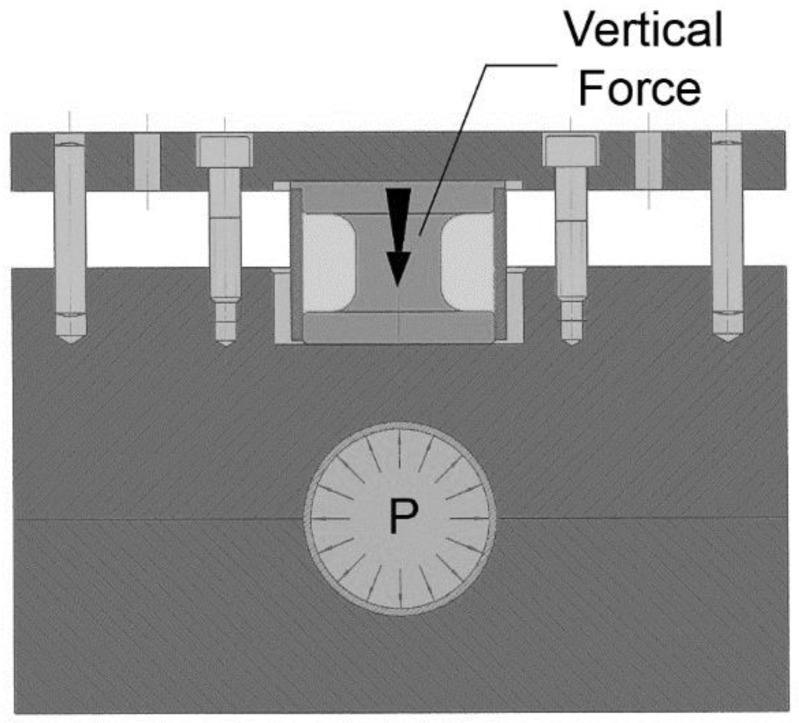
Cross-section of the tube-sliding tool.

**Figure 5 materials-15-05655-f005:**
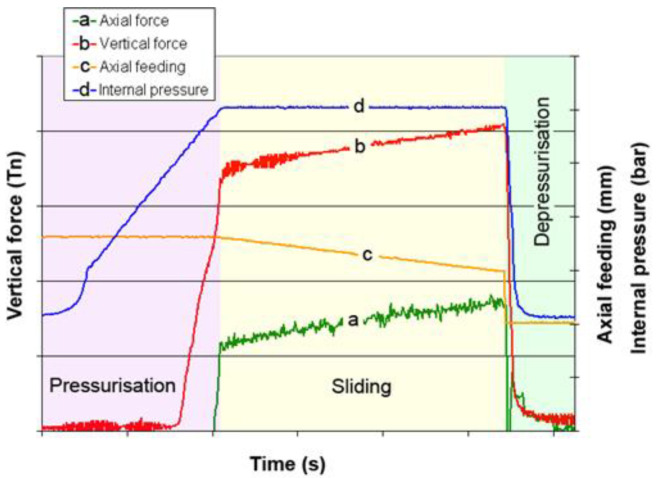
Different tool zones in tube hydroforming.

**Figure 6 materials-15-05655-f006:**
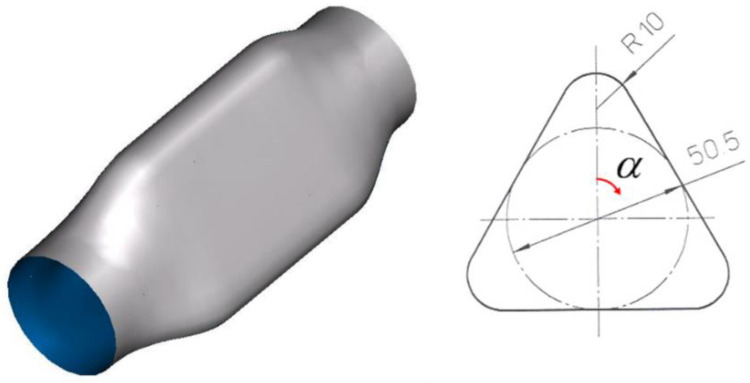
Selected triangular shape component.

**Figure 7 materials-15-05655-f007:**
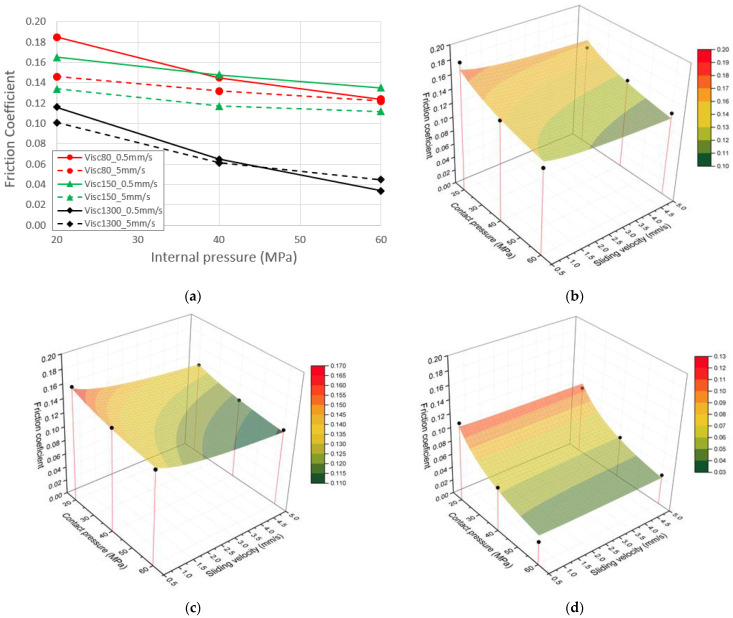
Tube-sliding friction test results. (**a**) Experimental results at different conditions, (**b**) fitted friction law for viscosity 80 lubricant, (**c**) fitted friction law for viscosity 150 lubricant and (**d**) fitted friction law for viscosity 1300 lubricant.

**Figure 8 materials-15-05655-f008:**
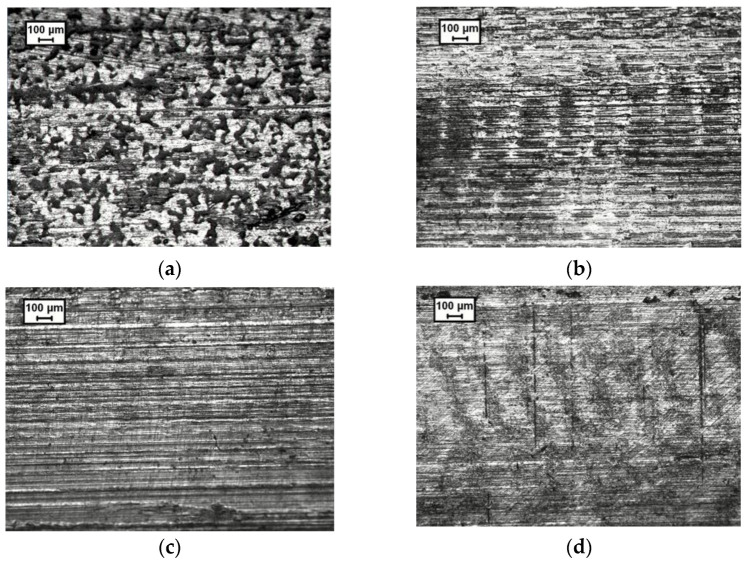
Tube outer surface micrographs. (**a**) Initial tube before testing, (**b**) micrograph of damaged zone for viscosity 80 lubricant, (**c**) micrograph of damaged zone for viscosity 150 lubricant and (**d**) micrograph of damaged zone for viscosity 1300 lubricant.

**Figure 9 materials-15-05655-f009:**
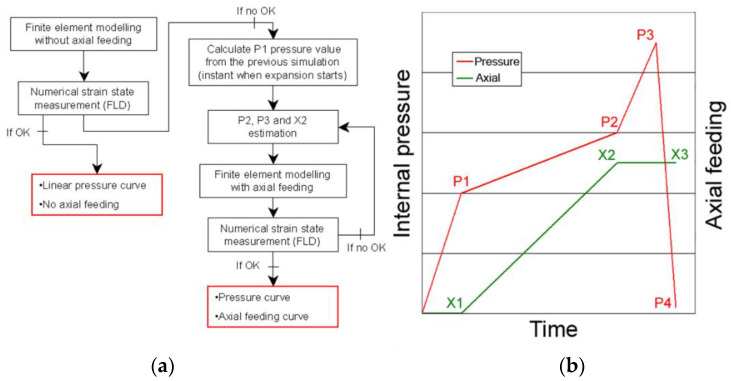
(**a**) Process design strategy and (**b**) schematic view of optimized pressure and axial feeding parameters.

**Figure 10 materials-15-05655-f010:**
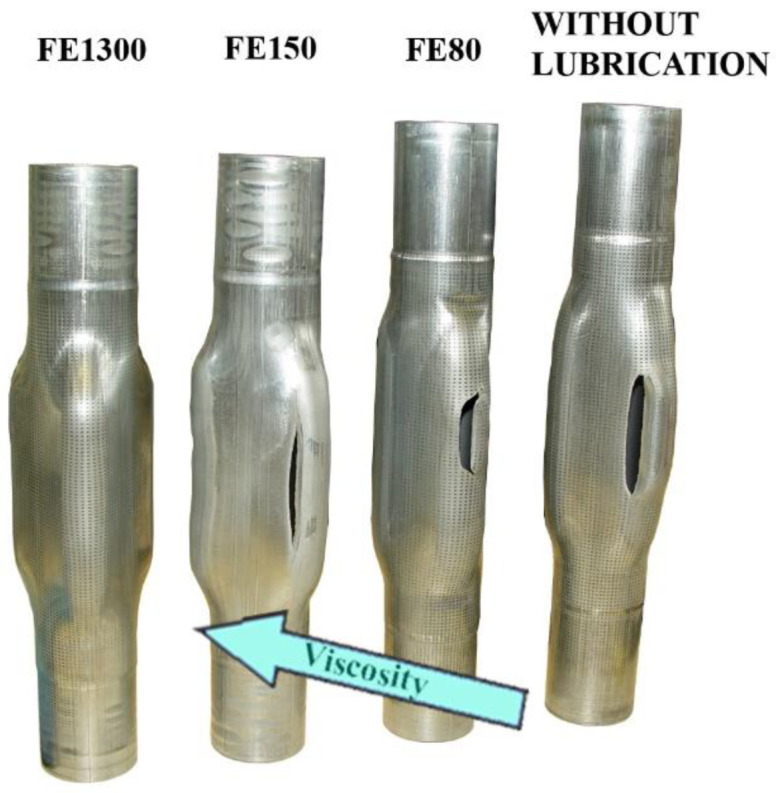
Hydroformed components with different viscosity lubricants and without lubrication. Viscosity increases from right to left.

**Figure 11 materials-15-05655-f011:**
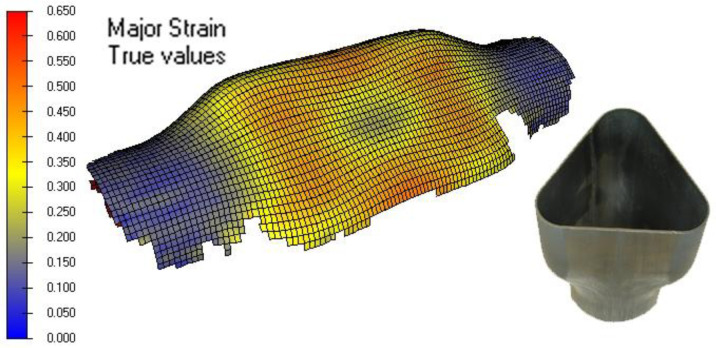
Major strain measured by photogrammetry.

**Figure 12 materials-15-05655-f012:**
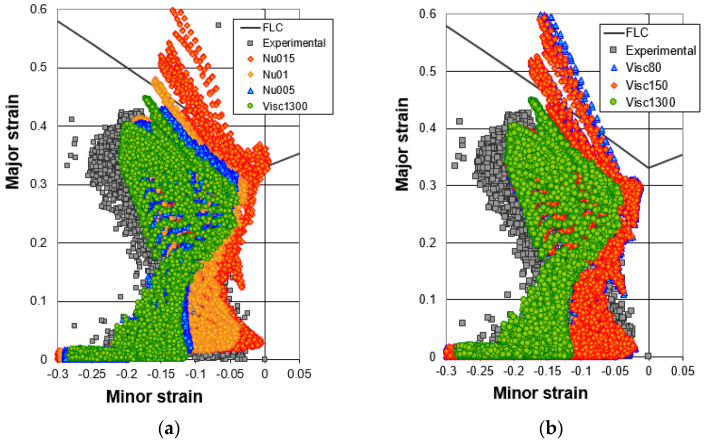
(**a**) Forming Limit Diagrams for constant friction coefficients, the high viscosity advanced friction model and (**b**) Forming Limit Diagrams for different advanced friction models and the experimental principal strain of hydroformed component.

**Figure 13 materials-15-05655-f013:**
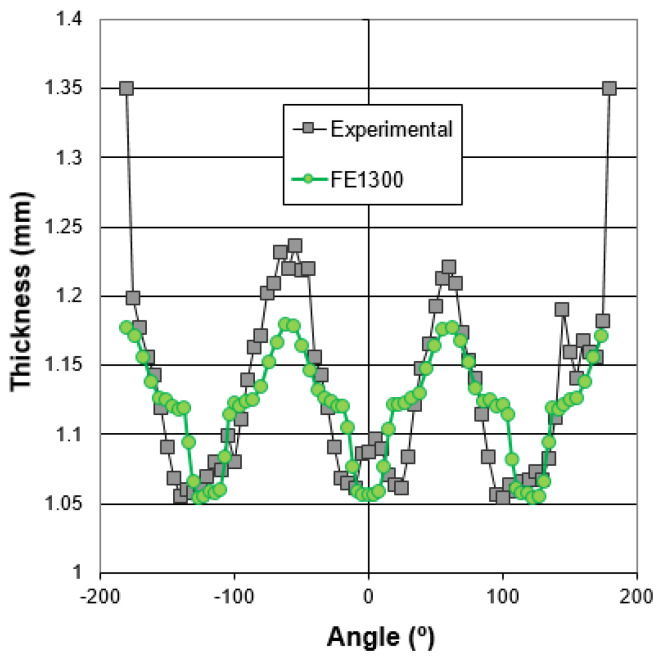
Final thickness distribution of the high viscosity lubricant experimental case and the numerical predictions.

**Figure 14 materials-15-05655-f014:**
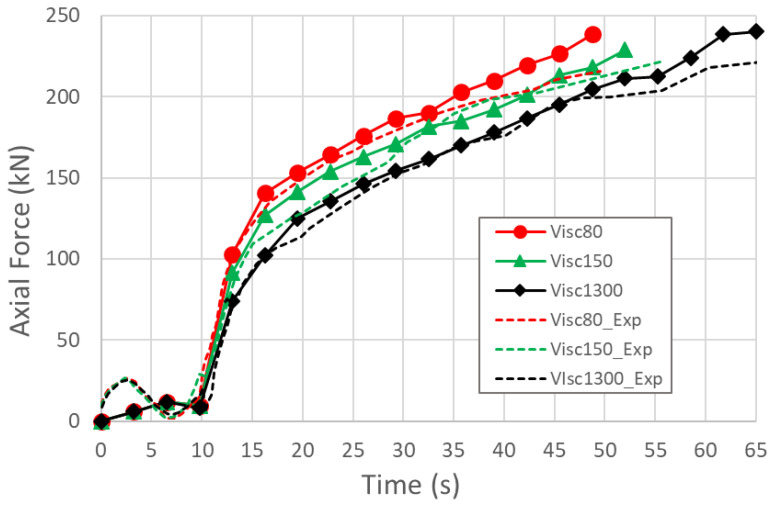
Experimental and numerical predictions of the axial feeding forces for the different lubricants.

**Table 1 materials-15-05655-t001:** Properties of tube material.

Mechanical Properties of DC03 Tube Material				
Yield Strength (MPa)		250		Contact extensometer
Ultimate Tensile Strength (MPa)		310		
Elongation at fracture (%)		>34		
Modulus of Elasticity (GPa)		205		GOM Aramis
Lankford coefficients (r_0_, r_45_, r_90_)		1.21, 0.91, 1.21		
SHS hardening law GOM Aramis small area		$\sigma =0.75\cdot \left[455\cdot {\left(\varepsilon_{p}+0.01\right)}^{0.2}\right]+0.25\cdot \left[650-\left(650-145\right)\cdot e^{-4.37\cdot \varepsilon_{p}^{0.69}}\right]$		
**Chemical Composition (%) Max. Weight Percentages**				
C	Mn	P	S	Fe
0.10	0.45	0.035	0.035	Rest

**Table 2 materials-15-05655-t002:** The 3D topographical parameters of the tube and tool surfaces.

3D Parameter	Average Tube	Standard Deviation	Average Tool	Standard Deviation
Sq (µm)	0.88	0.04	0.75	0.12
Sa (µm)	0.65	0.01	0.62	0.09
Sdq (°)	0.42	0.03	0.09	0.01
Sdr (%)	5.60	0.79	0.36	0.06
Vmp (*p* = 10%) [µm^3^/µm^2^]	3.03 × 10^−5^	4.84 × 10^−6^	0.03	0.01
Vvv (*p* = 80%) [µm^3^/µm^2^]	1.61 × 10^−4^	6.36 × 10^−6^	0.07	0.01

**Table 3 materials-15-05655-t003:** Properties of selected lubricants.

Lubricant	Cinematic Viscosity 40 °C (mm^2^/s)	Density (g/mL)	Ignition Temperature (°C)	Application
Lub 1	80	0.92	190	Tube bending and processing
Lub 2	150	0.92	193	Tube bending and processing
Lub 3	1300	0.94	195	Extreme pressure tube forming

**Table 4 materials-15-05655-t004:** Filzek law parameters for the different lubricants.

Lubricant	Cinematic Viscosity 40 °C (mm^2^/s)	*µ* _0_	*P*_0_ (MPa)	*n*	*k*	*V*_0_ (mm/s)	*R^2^*
Lub 1	80	0.175	19.5	0.75	0.0078	0.54	0.86
Lub 2	150	0.163	19.4	0.84	0.0123	0.55	0.98
Lub 3	1300	0.144	14.6	0.16	0.0012	0.25	0.95

## Data Availability

Not applicable.

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
