# Peer review of "Friction Modelling for Tube Hydroforming Processes—A Numerical and Experimental Study with Different Viscosity Lubricants"

_materials, 2022, doi:10.3390/ma15165655_

Round 1

Reviewer 1 Report

The manuscript entitled “Friction modelling for tube hydroforming processes. A numerical and experimental study with different viscosity lubricants” by Garcia et al. present an interesting study, in which the tube sliding test is used to evaluate contact pressure and sliding velocity dependent friction laws to improve tube hydroforming models. The authors tested their improved models experimentally by forming a triangular-shaped part using three different lubricants of different viscosities. The authors report good agreement and predictability.

Overall, this manuscript appears to be scientifically sound and of interest to the manufacturing community. The manuscript is well written, in particular the introduction presents a good background motivating the importance of this study to the general reader. The findings are well explained, and most figures support the narrative well. Below I have some comments to improve the manuscript further before final publication:

  • The authors point out the importance of temperature as a processing parameter which can vary, also locally within the tool, is recently evaluated by the community, and could significantly affect the outcome of the hydroforming process. For example:

“…currently, temperature dependency is being implemented as it has also a major effect on friction. In this work, three lubricants having different viscosity have been characterized using the tube sliding test.” (Lines 16 to 18)

I understand that temperature is not the focus of this study, which is fine. After mentioning this aspect in both the abstract and introduction, it would be good if the authors could comment on this matter in the discussion section. In particular, the viscosity of the lubricants is presumably highly temperature dependent. Can the authors estimate how the temperature varies at the different hydroforming tool zones? In particular, does straining the material at different rates at different points of the part, as well as friction lead to significant temperature changes? To what degree are these changes expected to affect the results presented here?

  • Line 141: “a straight tube is continuously upset in a closed die”. It is unclear what “upset” means here.

  • Line 224: “80 and 150 mm2/s respectively and they are typically used in the tube bending and forming of steel components. The third lubricant has a high viscosity, 1300 mm2/s,…”

To my understanding, “mm^2/sec" is not the correct unit for the viscosity. The traditional unit for viscosity is Poise, the SI unit for viscosity is Pa.sec. Converting Pa.sec in the below dimensional analysis, I cannot obtain mm^2/sec as a unit. The authors should clarify or correct this:

10 Poise = 1 Pa sec = 1 N m^-2 sec = 1 kg m sec^-2 m^-2 sec = 1 kg m^-1 sec^-1

  • Line 227: “All the lubricants have been produced by the same supplier”. I cannot find any supplier information. This needs to be provided.

  • Figure 10: The arrow is confusing. I believe the authors are trying to show that the viscosity increases from right to left along the arrow as an axis. But instead, it looks like the arrow is pointing at the FE1300 part. Maybe the authors can use a different style arrow and place it horizontally below all parts to make this axis clearer.

  • Line 423 should read “etched” not “edged”.

  • Figure 12 is confusing.

Firstly, to aid the general reader the authors should include a couple of sentences to explain what the Forming Limit Diagrams show and what can be learned from such diagrams in general. Not every reader will be familiar with this.

Secondly, there are two panels, and it is not immediately clear to me what the difference between the left and the right panel is. I do see that the legend entries are partially different. Best to label panels as a) and b) and explain this in the caption.

Thirdly, there are a lot of overlapping data points in these plots. This makes it difficult to comprehend and compare the distributions of these data.

  • In lines 429 to 437 the authors draw conclusion about the deformation and friction behavior from Fig. 12. It would be good if the authors could explain better to the reader what specific features or differences in or between these plots and the data lead to these conclusions. Otherwise, the reader has trouble following the line of argument.

  • In Figure 14, the axial force is plotted in units of “tn”. What is this unit?

Author Response

Our answers are attached as a word file

Reviewer 2 Report

The manuscript entitled „Friction modelling for tube hydroforming processes. A numerical and experimental study with different viscosity lubricants“ is scientifically written at a good level bringing original results interesting for practical use. Results showed that the correct choice of lubricant has a significant impact on the friction which can prevent damage during the hydroforming process.

However, the article does not have an appropriate stylistic form, structure and arrangement of the acquired knowledge typical for this type of publications. The main major comments are given below:

  1. Chapters 1 and 2 resemble chapters in books. These chapters should be shortened and compressed to compact scientific overview.
  2. It is also not entirely appropriate to combine “Discussion” and “Conclusions” chapters into one common chapter. Conclusions should be self-supporting, clear and compact.
  3. It would also be suitable to have a separate chapter, for example entitles “Experimental ad computational procedures”, where the reader has the opportunity to easily and clearly find all the materials and methods used including measurement parameters, or calculation parameters in one place. It is not customary to list them at the beginning of the particular chapters (subchapters in Chapters 3 and 4).

The other minor comments:

  1. Fig. 4 – it is not clear, whether the image is taken from the literature or created by the authors. If it is redrawn from the literature, a reference must be added.
  2. Fig. 2 is not necessary, rather it fits into the diploma thesis.
  3. Fig. 7 – authors should specify (at least in the text) the position in the sample, in which the micrograph was taken. Besides that, the designation of particular micrographs (a, b, c and d) is hardly visible to the reader.
  4. The discussion could include a sentence explaining why the values of “k” and “n” are highest for Lubricant 2.
  5. Figures 6, 13 and 14 – the values of the measured points are connected linearly. However, such a course may not correspond to the real course. This is mostly evident in Fig. 6a, where only three measured points corresponding to 20, 40 and 60 MPa are interconnected with lines. It would be more suitable to use an approximation function, or not to interconnect the points at all.
  6. Table 1 – authors should specify the atomic or weight percentages.
  7. Table 2 – standard deviations values should have the same decimal places as the average value.

Author Response

Our answers are attached as a word file

Round 2

Reviewer 2 Report

I accept all arguments and proposed changes from the authors. I recommend the article for publication.

I suggest one minor change - it is at the author's discretion (Question/comment 6):

Figures 6a - I suggest replacing solid lines for 0.5 mm/s with a dotted line.